# Generating Adversarial Computer Programs using Optimized Obfuscations

**Shashank Srikant**[1]    **Sijia Liu**[2,3]    **Tamara Mitrovska**[1]    **Shiyu Chang**[2]
**Quanfu Fan**[2]    **Gaoyuan Zhang**[2]    **Una-May O'Reilly**[1]

[1]CSAIL, MIT    [2]MIT-IBM Watson AI Lab    [3]Michigan State University
shash@mit.edu, liusiji5@msu.edu, unamay@csail.mit.edu

## Abstract

Machine learning (ML) models that learn and predict properties of computer programs are increasingly being adopted and deployed. In this work, we investigate principled ways to adversarially perturb a computer program to fool such learned models, and thus determine their adversarial robustness. We use program obfuscations, which have conventionally been used to avoid attempts at reverse engineering programs, as adversarial perturbations. These perturbations modify programs in ways that do not alter their functionality but can be crafted to deceive an ML model when making a decision. We provide a general formulation for an adversarial program that allows applying multiple obfuscation transformations to a program in any language. We develop first-order optimization algorithms to efficiently determine two key aspects – which parts of the program to transform, and what transformations to use. We show that it is important to optimize both these aspects to generate the best adversarially perturbed program. Due to the discrete nature of this problem, we also propose using randomized smoothing to improve the attack loss landscape to ease optimization. We evaluate our work on Python and Java programs on the problem of program summarization.[1] We show that our best attack proposal achieves a $52\%$ improvement over a state-of-the-art attack generation approach for programs trained on a SEQ2SEQ model. We further show that our formulation is better at training models that are robust to adversarial attacks.

## 1 Introduction

Machine learning (ML) models are increasingly being used for software engineering tasks. Applications such as refactoring programs, auto-completing them in editors, and synthesizing GUI code have benefited from ML models trained on large repositories of programs, sourced from popular websites like GitHub (Allamanis et al., 2018). They have also been adopted to reason about and assess programs (Srikant & Aggarwal, 2014; Si et al., 2018), find and fix bugs (Gupta et al., 2017; Pradel & Sen, 2018), detect malware and vulnerabilities in them (Li et al., 2018; Zhou et al., 2019) *etc*. thus complementing traditional program analysis tools. As these models continue to be adopted for such applications, it is important to understand how robust they are to *adversarial attacks*. Such attacks can have adverse consequences, particularly in settings such as security (Zhou et al., 2019) and compliance automation (Pedersen, 2010). For example, an attacker could craft changes in malicious programs in a way which forces a model to incorrectly classify them as being benign, or make changes to pass off code which is licensed as open-source in an organization's proprietary code-base.

Adversarially perturbing a program should achieve two goals – a trained model should flip its decision when provided with the perturbed version of the program, and second, the perturbation should be *imperceivable*. Adversarial attacks have mainly been considered in image classification (Goodfellow et al., 2014; Carlini & Wagner, 2017; Madry et al., 2018), where calculated minor changes made to pixels of an image are enough to satisfy the imperceptibility requirement. Such changes escape a human's attention by making the image look the same as before perturbing it, while modifying the underlying representation enough to flip a classifier's decision. However, programs demand a stricter imperceptibility requirement – not only should the changes avoid human attention, but the changed program should also importantly functionally behave the same as the unperturbed program.

---

[1]Source code: https://github.com/ALFA-group/adversarial-code-generation

*Program obfuscations* provide the agency to implement one such set of imperceivable changes in programs. Obfuscating computer programs have long been used as a way to avoid attempts at reverse-engineering them. They transform a program in a way that only hampers humans' comprehension of parts of the program, while retaining its original semantics and functionality. For example, one common obfuscation operation is to rename variables in an attempt to hide the program's intent from a reader. Renaming a variable `sum` in the program statement `int sum = 0` to `int xyz = 0` neither alters how a compiler analyzes this variable nor changes any computations or states in the program; it only hampers our understanding of this variable's role in the program. Modifying a very small number of such aspects of a program marginally affects how we comprehend it, thus providing a way to produce changes imperceivable to both humans and a compiler. In this work, we view adversarial perturbations to programs as a special case of applying obfuscation transformations to them.

Having identified a set of candidate transformations which produce imperceivable changes, a specific subset needs to be chosen in a way which would make the transformed program adversarial. Recent attempts (Yefet et al., 2019; Ramakrishnan et al., 2020; Bielik & Vechev, 2020) which came closest to addressing this problem did not offer any rigorous formulation. They recommended using a variety of transformations without presenting any principled approach to selecting an optimal subset of transformations. We present a formulation which when solved provides the exact location to transform as well as a transformation to apply at the location. Figure 1 illustrates this. A randomly selected local-variable (`name`) when replaced by the name `virtualname`, which is generated by the state-of-the-art attack generation algorithm for programs

**Unperturbed**
```
def __setitem__(self, name, val):
    name, val = forbid_multi_line_headers(name, val, self.encoding)
    MIMEText.__setitem__(self, name, val)
                                        Prediction: set item
```

**Random site-selection; Optimal site-perturbation** (Ramakrishnan et al., 2020)
```
def __setitem__(self, name, val):
    virtualname, val = forbid_multi_line_headers(name, val, self.encoding)
    MIMEText.__setitem__(self, virtualname, val)
                                        Prediction: set item
```

**Optimal site-selection; Optimal site-perturbation** (This work)
```
def __setitem__(self, qisrc, val):
    name, val = forbid_multi_line_headers(qisrc, val, self.encoding)
    MIMEText.__setitem__(self, name, val)
                                        Prediction: write
```

Figure 1: The advantage of our formulation when compared to the state-of-the-art.

(Ramakrishnan et al., 2020), is unable to fool a program summarizer (which predicts `set item`) unless our proposed site optimization is applied. We provide a detailed comparison in Section 2. In our work, we make the following key contributions –

- We identify two problems central to defining an adversarial program – identifying the sites in a program to apply perturbations on, and the specific perturbations to apply on the selected sites. These perturbations are involve replacing existing tokens or inserting new ones.
- We provide a general mathematical formulation of a perturbed program that models site locations and the perturbation choice for each location. It is independent of programming languages and the task on which a model is trained, while seamlessly modeling the application of multiple transformations to the program.
- We propose a set of first-order optimization algorithms to solve our proposed formulation efficiently, resulting in a differentiable generator for adversarial programs. We further propose a randomized smoothing algorithm to achieve improved optimization performance.
- Our approach demonstrates a 1.5x increase in the attack success rate over the state-of-the-art attack generation algorithm (Ramakrishnan et al., 2020) on large datasets of Python and Java programs.
- We further show that our formulation provides better robustness against adversarial attacks compared to the state-of-the-art when used in training an ML model.

## 2 RELATED WORK

Due to a large body of literature on adversarial attacks in general, we focus on related works in the domain of computer programs. Wang & Christodorescu (2019), Quiring et al. (2019), Rabin et al. (2020), and Pierazzi et al. (2020) identify obfuscation transformations as potential adversarial examples. They do not, however, find an optimal set of transformations to deceive a downstream model. Liu et al. (2017) provide a stochastic optimization formulation to obfuscate programs optimally by maximizing its impact on an *obscurity language model* (OLM). However, they do not address the problem of adversarial robustness of ML models of programs, and their formulation is only to find the right sequence of transformations which increases their OLM's perplexity. They use an MCMC-based search to find the best sequence.

Yefet et al. (2019) propose perturbing programs by replacing local variables, and inserting print statements with replaceable string arguments. They find optimal replacements using a first-order optimization method, similar to Balog et al. (2016) and HotFlip (Ebrahimi et al., 2017). This is

Figure 2: (a) A sample program $\mathcal{P}$ containing a function foo (b) $\mathcal{P}$ contains five *sites* which can be transformed - two *replace* sites corresponding to local variables b and r , and three *insert* sites at locations I1, I2, I3. $\Omega$ is a vocabulary of tokens which can be used for the transformations. (c) This is a *perturbed program* with the tokens world and set from $\Omega$ used to replace tokens b and at location I3. These transformations do not change the original functionality of $\mathcal{P}$, but cause an incorrect prediction delete (d) Examples of two site selection vectors $\mathbf{z}^i$, $\mathbf{z}^{ii}$ selecting different components. $\mathbf{z}_i = 1$ for a location $i$ signifies that the $i$th token in $\mathcal{P}$ is selected to be optimally transformed. $\mathbf{z}^i$ corresponds to the perturbed program in (c).

an improvement over Zhang et al. (2020), who use the Metropolis-Hastings algorithm to find an optimal replacement for variable names. Bielik & Vechev (2020) propose a robust training strategy which trains a model to abstain from deciding when uncertain if an input program is adversarially perturbed. The transformation space they consider is small, which they search through greedily. Moreover, their solution is designed to reason over a limited context of the program (predicting variable types), and is non-trivial to extend to applications such as program summarization (explored in this work) which requires reasoning over an entire program.

Ramakrishnan et al. (2020) extend the work by Yefet et al. (2019) and is most relevant to what we propose in this work. They experiment with a larger set of transformations and propose a standard min-max formulation to adversarially train robust models. Their inner-maximizer, which generates adversarial programs, models multiple transformations applied to a program in contrast to Yefet et al. (2019). However, they do not propose any principled way to solve the problem of choosing between multiple program transformations. They randomly select transformation operations to apply, and then randomly select locations in the program to apply those transformations on.

We instead show that optimizing for locations alone improves the attack performance. Further, we propose a joint optimization problem of finding the optimal location and optimal transformation, only the latter of which Ramakrishnan et al. (2020) (and Yefet et al. (2019)) address in a principled manner. Although formally unpublished at the time of preparing this work, we compare our experiments to Ramakrishnan et al. (2020), the state-of-the-art in evaluating and defending against adversarial attacks on models for programs, and contrast the advantages of our formulation.

## 3    PROGRAM OBFUSCATIONS AS ADVERSARIAL PERTURBATIONS

In this section, we formalize program obfuscation operations, and show how generating adversarial programs can be cast as a constrained combinatorial optimization problem.

**Program obfuscations.** We view obfuscation transformations made to programs as adversarial perturbations which can affect a downstream ML/DL model like a malware classifier or a program summarizer. While a variety of such obfuscation transformations exist for programs in general (see section 2A, Liu et al. (2017)), we consider two broad classes – *replace* and *insert* transformations. In *replace* transformations, existing program constructs are replaced with variants which decrease readability. For example, replacing a variable's name, a function parameter's name, or an object field's name does not affect the semantics of the program in any way. These names in any program exclusively aid human comprehension, and thus serve as three *replace* transformations. In *insert* transformations, we insert new statements to the program which are unrelated to the code it is inserted around, thereby obfuscating its original intent. For example, including a print statement with an arbitrary string argument does not change the semantics of the program in any way.

Our goal hence is to introduce a systematic way to transform a program with *insert* or *replace* transformations such that a trained model misclassifies a program $\mathcal{P}$ that it originally classified correctly.

***Site-selection* and *Site-perturbation* – Towards defining adversarial programs.** Before we formally define an adversarial program, we highlight the key factors which need to be considered in our formulation through the example program introduced in Figure 2.

Consider applying the following two obfuscation transformations on the example program $\mathcal{P}$ in Figure 2.a – replacing local variable names (a *replace* transform), and inserting `print` statements (an *insert* transform). The two local variables `b` and `r` in $\mathcal{P}$ are potential candidates where the replace transform can be applied, while a `print` statement can potentially be inserted at the three locations `I1`, `I2`, `I3` (highlighted in Figure 2.b). We notate these choices in a program as *sites*–locations in a program where a unique transformation can be applied.

Thus, in order to *adversarially perturb* $\mathcal{P}$, we identify two important questions that need to be addressed. First, which sites in a program should be transformed? Of the $n$ sites in a program, if we are allowed to choose at most $k$ sites, which set of $\leq k$ sites would have the highest impact on the downstream model's performance? We identify this as the ***site-selection problem***, where the constraint $k$ is the ***perturbation strength*** of an attacker. Second, what tokens should be inserted/replaced at the $k$ selected sites? Once we pick $k$ sites, we still have to determine the best choice of tokens to replace/insert at those sites which would have the highest impact on the downstream model. We refer to this as the ***site-perturbation problem***.

**Mathematical formulation.** In what follows, we propose a general and rigorous formulation of adversarial programs. Let $\mathcal{P}$ denote a *benign program* which consists of a series of $n$ tokens $\{\mathcal{P}_i\}_{i=1}^n$ in the source code domain. For example, the program in Figure 2.a, when read from top to bottom and left to right, forms a series of $n = 12$ tokens $\{\text{def}, \text{b}, ..., \text{r}, +, 5\}$. We ignore white spaces and other delimiters when tokenizing. Each $\mathcal{P}_i \in \{0,1\}^{|\Omega|}$ here is considered a one-hot vector of length $|\Omega|$, where $\Omega$ is a vocabulary of tokens. Let $\mathcal{P}'$ define a *perturbed program* (with respect to $\mathcal{P}$) created by solving the *site-selection* and *site-perturbation* problems, which use the vocabulary $\Omega$ to find an optimal replacement. Since our formulation is agnostic to the type of transformation, *perturbation* in the remainder of this section refers to both *replace* and *insert* transforms. In our work, we use a shared vocabulary $\Omega$ to select transforms from both these classes. In practice, we can also assign a unique vocabulary to each transformation we define.

To formalize the *site-selection* problem, we introduce a vector of boolean variables $\mathbf{z} \in \{0,1\}^n$ to indicate whether or not a site is selected for perturbation. If $z_i = 1$ then the $i$th site (namely, $\mathcal{P}_i$) is perturbed. If there exist multiple occurrences of a token in the program, then all such sites are marked 1. For example, in Figure 2.d, if the site corresponding to local variable `b` is selected, then both indices of its occurrences, $z_3, z_9$ are marked as 1 as shown in $z^i$. Moreover, the number of perturbed sites, namely, $\mathbf{1}^T \mathbf{z} \leq k$ provides a means of measuring *perturbation strength*. For example, $k = 1$ is the minimum perturbation possible, where only one site is allowed to be perturbed. To define *site-perturbation*, we introduce a one-hot vector $\mathbf{u}_i \in \{0,1\}^{|\Omega|}$ to encode the selection of a token from $\Omega$ which would serve as the insert/replace token for a chosen transformation at a chosen site. If the $j^{\text{th}}$ entry $[\mathbf{u}_i]_j = 1$ and $z_i = 1$, then the $j$th token in $\Omega$ is used as the obfuscation transformation applied at the site $i$ (namely, to perturb $\mathcal{P}_i$). We also have the constraint $\mathbf{1}^T \mathbf{u}_i = 1$, implying that only one perturbation is performed at $\mathcal{P}_i$. Let vector $\mathbf{u} \in \{0,1\}^{n \times |\Omega|}$ denote $n$ different $\mathbf{u}_i$ vectors, one for each token $i$ in $\mathcal{P}$.

Using the above formulations for *site-selection*, *site-perturbation* and *perturbation strength*, the *perturbed program* $\mathcal{P}'$ can then be defined as

$$\mathcal{P}' = (\mathbf{1} - \mathbf{z}) \cdot \mathcal{P} + \mathbf{z} \cdot \mathbf{u}, \text{ where } \mathbf{1}^T \mathbf{z} \leq k, \mathbf{z} \in \{0,1\}^n, \mathbf{1}^T \mathbf{u}_i = 1, \mathbf{u}_i \in \{0,1\}^{|\Omega|}, \forall i, \quad (1)$$

where $\cdot$ denotes the element-column wise product.

The adversarial effect of $\mathcal{P}'$ is then measured by passing it as input to a downstream ML/DL model $\boldsymbol{\theta}$ and seeing if it successfully manages to fool it.

Generating a successful adversarial program is then formulated as the optimization problem,

$$\begin{aligned} \underset{\mathbf{z},\mathbf{u}}{\text{minimize}} \quad & \ell_{\text{attack}}((\mathbf{1} - \mathbf{z}) \cdot \mathcal{P} + \mathbf{z} \cdot \mathbf{u}; \mathcal{P}, \boldsymbol{\theta}) \\ \text{subject to} \quad & \text{constraints in (1),} \end{aligned} \quad (2)$$

where $\ell_{\text{attack}}$ denotes an attack loss. In this work, we specify $\ell_{\text{attack}}$ as the cross-entropy loss on the predicted output evaluated at $\mathcal{P}'$ in an untargeted setting (namely, without specifying the prediction label targeted by an adversary) (Ramakrishnan et al., 2020). One can also consider other specifications of $\ell_{\text{attack}}$, e.g., C&W untargeted and targeted attack losses (Carlini & Wagner, 2017).

## 4 ADVERSARIAL PROGRAM GENERATION VIA FIRST-ORDER OPTIMIZATION

Solving problem (2) is not trivial because of its combinatorial nature (namely, the presence of boolean variables), the presence of a bi-linear objective term (namely, $\mathbf{z} \cdot \mathbf{u}$), as well as the presence of multiple constraints. To address this, we present a projected gradient descent (PGD) based joint optimization solver (JO) and propose alternates which promise better empirical performance.

**PGD as a joint optimization (JO) solver.** PGD has been shown to be one of the most effective attack generation methods to fool image classification models (Madry et al., 2018). Prior to applying PGD, we instantiate (2) into a feasible version by relaxing boolean constraints to their convex hulls,

$$
\begin{aligned}
\underset{\mathbf{z},\mathbf{u}}{\text{minimize}} \quad & \ell_{\text{attack}}(\mathbf{z}, \mathbf{u}) \\
\text{subject to} \quad & \mathbf{1}^T \mathbf{z} \leq k, \ \mathbf{z} \in [0,1]^n, \ \mathbf{1}^T \mathbf{u}_i = 1, \ \mathbf{u}_i \in [0,1]^{|\Omega|}, \ \forall i,
\end{aligned}
\tag{3}
$$

where for ease of notation, the attack loss in (2) is denoted by $\ell_{\text{attack}}(\mathbf{z}, \mathbf{u})$. The continuous relaxation of binary variables in (3) is a commonly used trick in combinatorial optimization to boost the stability of learning procedures in practice (Boyd et al., 2004). Once the continuous optimization problem (3) is solved, a hard thresholding operation or a randomized sampling method (which regards $\mathbf{z}$ and $\mathbf{u}$ as probability vectors with elements drawn from a Bernoulli distribution) can be called to map a continuous solution to its discrete domain (Blum & Roli, 2003). We use the randomized sampling method in our experiments.

The PGD algorithm is then given by

$$
\{\mathbf{z}^{(t)}, \mathbf{u}^{(t)}\} = \{\mathbf{z}^{(t-1)}, \mathbf{u}^{(t-1)}\} - \alpha\{\nabla_{\mathbf{z}}\ell_{\text{attack}}(\mathbf{z}^{(t-1)}, \mathbf{u}^{(t-1)}), \nabla_{\mathbf{u}}\ell_{\text{attack}}(\mathbf{z}^{(t-1)}, \mathbf{u}^{(t-1)})\}
\tag{4}
$$

$$
\{\mathbf{z}^{(t)}, \mathbf{u}^{(t)}\} = \text{Proj}(\{\mathbf{z}^{(t)}, \mathbf{u}^{(t)}\}),
\tag{5}
$$

where $t$ denotes PGD iterations, $\mathbf{z}^{(0)}$ and $\mathbf{u}^{(0)}$ are given initial points, $\alpha > 0$ is a learning rate, $\nabla_{\mathbf{z}}$ denotes the first-order derivative operation w.r.t. the variable $\mathbf{z}$, and $\text{Proj}$ represents the projection operation w.r.t. the constraints of (3).

The projection step involves solving for $\mathbf{z}$ and $\mathbf{u}_i$ simultaneously in a complex convex problem. See Equation 9 in Appendix A for details.

A key insight is that the complex projection problem (9) can equivalently be *decomposed* into a sequence of sub-problems owing to the separability of the constraints w.r.t. $\mathbf{z}$ and $\{\mathbf{u}_i\}$. The two sub-problems are –

$$
\underset{\mathbf{z}}{\text{minimize}} \quad \|\mathbf{z} - \mathbf{z}^{(t)}\|_2^2 \qquad \text{and} \qquad \underset{\mathbf{u}_i}{\text{minimize}} \quad \|\mathbf{u}_i - \mathbf{u}_i^{(t)}\|_2^2 \qquad \forall i.
$$
$$
\text{subject to} \quad \mathbf{1}^T\mathbf{z} \leq k, \ \mathbf{z} \in [0,1]^n, \qquad \text{subject to} \quad \mathbf{1}^T\mathbf{u}_i = 1, \ \mathbf{u}_i \in [0,1]^{|\Omega|},
\tag{6}
$$

The above subproblems w.r.t. $\mathbf{z}$ and $\mathbf{u}_i$ can optimally be solved by using a bisection method that finds the root of a scalar equation. We provide details of a closed-form solution and its corresponding proof in Appendix A. We use this decomposition and solutions to design an alternating optimizer, which we discuss next.

**Alternating optimization (AO) for fast attack generation.** While JO provides an approach to solve the unified formulation in (2), it suffers from the problem of getting trapped at a poor local optima despite attaining stationarity (Ghadimi et al., 2016). We propose using AO (Bezdek & Hathaway, 2003) which allows the loss landscape to be explored more aggressively, thus leading to better empirical convergence and optimality (see Figure 4a).

AO solves problem (2) one variable at a time – first, by optimizing the site selection variable $\mathbf{z}$ keeping the site perturbation variable $\mathbf{u}$ fixed, and then optimizing $\mathbf{u}$ keeping $\mathbf{z}$ fixed. That is,

$$
\mathbf{z}^{(t)} = \underset{\mathbf{1}^T\mathbf{z} \leq k, \ \mathbf{z} \in [0,1]^n}{\arg\min} \ell_{\text{attack}}(\mathbf{z}, \mathbf{u}^{(t-1)}) \quad \text{and} \quad \mathbf{u}_i^{(t)} = \underset{\mathbf{1}^T\mathbf{u}_i = 1, \ \mathbf{u}_i \in [0,1]^{|\Omega|}}{\arg\min} \ell_{\text{attack}}(\mathbf{z}^{(t)}, \mathbf{u}) \ \forall i.
\tag{7}
$$

We can use PGD, as described in (6), to similarly solve each of $\mathbf{z}$ and $\mathbf{u}$ separately in the two alternating steps. Computationally, AO is expensive than JO by a factor of 2, since we need two iterations to cover all the variables which JO covers in a single iteration. However, in our experiments, we find AO to converge faster. The decoupling in AO also eases implementation, and provides the flexibility to set a different number of iterations for the $u$-step and the $z$-step within one iteration of AO. We also remark that the AO setup in (7) can be specified in other forms, *e.g.* alternating direction method of multipliers (ADMM) (Boyd et al., 2011). However, such methods use an involved alternating scheme to solve problem (2). We defer evaluating these options to future work.

**Randomized smoothing (RS) to improve generating adversarial programs.** In our experiments, we noticed that the loss landscape of generating adversarial program is not smooth (Figure 3). This motivated us to explore surrogate loss functions which could smoothen it out. In our work, we employ a convolution-based RS technique (Duchi et al., 2012) to circumvent the optimization difficulty induced by the non-smoothness of the attack loss $\ell_{\text{attack}}$. We eventually obtain a smoothing loss $\ell_{\text{smooth}}$:

$$\ell_{\text{smooth}}(\mathbf{z}, \mathbf{u}) = \mathbb{E}_{\boldsymbol{\xi}, \boldsymbol{\tau}}[\ell_{\text{attack}}(\mathbf{z} + \mu\boldsymbol{\xi}, \mathbf{u} + \mu\boldsymbol{\tau})], \qquad (8)$$

where $\boldsymbol{\xi}$ and $\boldsymbol{\tau}$ are random samples drawn from the uniform distribution within the unit Euclidean ball, and $\mu > 0$ is a small smoothing parameter (set to $0.01$ in our experiments).

The rationale behind RS (8) is that the convolution of two functions (smooth probability density function and non-smooth attack loss) is at least as smooth as the smoothest of the two original functions. The advantage of such a formulation is that it is independent of the loss function, downstream model, and the optimization solver chosen for a problem. We evaluate RS on both AO and JO. In practice, we consider an empirical Monte Carlo approximation of (8), $\ell_{\text{smooth}}(\mathbf{z}, \mathbf{u}) = \sum_{j=1}^{m}[\ell_{\text{attack}}(\mathbf{z} + \mu\boldsymbol{\xi}_j, \mathbf{u} + \mu\boldsymbol{\tau}_j)]$. We set $m = 10$ in our experiments to save on computation time. We also find that smoothing the site perturbation variable $u$ contributes the most to improving attack performance. We hence perturb only $u$ to further save computation time.

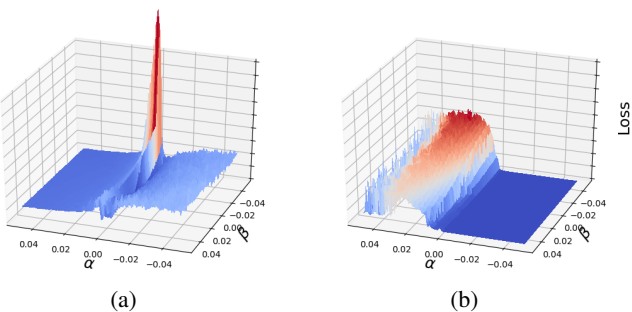

(a)          (b)

Figure 3: The original loss landscape for a sample program (3a). Randomized smoothing produces a flatter and smoother loss landscape (3b). We plot the loss along the space determined by the vector $(\alpha.\text{sgn}(\nabla_x f(x)) + \beta.\text{Rademacher}(0.5))$ for $\alpha, \beta \in [-0.05, 0.05]$ (Engstrom et al., 2018)

## 5    EXPERIMENTS & RESULTS

We begin by discussing the following aspects of our experiment setup – the classification task, the dataset and model we evaluate on, and the evaluation metrics we use.

**Task, Transformations, Dataset.** We evaluate our formulation of generating optimal adversarial programs on the problem of program summarization, first introduced by Allamanis et al. (2016). Summarizing a function in a program involves predicting its name, which is usually indicative of its intent. We use this benchmark to test whether our adversarially perturbed program, which retains the functionality of the original program, can force a trained summarizer to predict an incorrect function name. We evaluate this on a well maintained dataset of roughly 150K Python programs(Raychev et al., 2016) and 700K Java programs (Alon et al., 2018). They are pre-processed into functions, and each function is provided as input to an ML model. The name of the function is omitted from the input. The ML model predicts a sequence of tokens as the function name. We evaluate our work on six transformations (4 *replace* and 2 *insert* transformations); see Appendix B for details on these transformations. The results and analysis that follow pertains to the case when any of these six transformations can be used as a valid perturbation, and the optimization selects which to pick and apply based on the *perturbation strength* $k$. This is the same setting employed in the baseline (Ramakrishnan et al., 2020).

**Model.** We evaluate a trained SEQ2SEQ model. It takes program tokens as input, and generates a sequence of tokens representing its function name. We note that our formulation is independent of the learning model, and can be evaluated on any model for any task. The SEQ2SEQ model is trained and validated on $90\%$ of the data while tested on the remaining $10\%$. It is optimized using the cross-entropy loss function.

CODE2SEQ (Alon et al., 2018) is another model which has been evaluated on the task of program summarization. Its architecture is similar to that of SEQ2SEQ and contains two encoders - one which encodes tokens, while another which encodes AST paths. The model when trained only on tokens performs similar to a model trained on both tokens and paths (Table 3, Alon et al. (2018)). Thus adversarial changes made to tokens, as accommodated by our formulation, should have a high impact on the model's output. Owing to the similarity in these architectures, and since our

| Method | $k = 1$ site | | | | $k = 5$ sites | | | |
|---|---|---|---|---|---|---|---|---|
| | ASR | | F1 | | ASR | | F1 | |
| No attack | 0.00 | | 100.00 | | 0.00 | | 100.00 | |
| Random replace | 0.00 | | 100.00 | | 0.00 | | 100.00 | |
| Baseline* | 19.87 | | 78.18 | | 37.50 | | 59.54 | |
| AO | 23.16 | +3.29 ▲ | 74.78 | -3.40 ▲ | 43.53 | +6.03 ▲ | 53.75 | -5.79 ▲ |
| JO | 23.32 | +3.45 ▲ | 74.56 | -3.62 ▲ | 41.95 | +4.45 ▲ | 56.06 | -3.48 ▲ |
| AO + RS | 30.25 | +10.38 ▲ | 69.52 | -8.66 ▲ | 51.68 | +14.18 ▲ | 47.92 | -11.62 ▲ |
| JO + RS | 23.95 | +4.08 ▲ | 74.24 | -3.94 ▲ | 48.70 | +11.20 ▲ | 51.55 | -7.99 ▲ |

Table 1: Our work solves two key problems to find optimal adversarial perturbations – *site-selection* and *site-perturbation*. The Baseline method refers to (Ramakrishnan et al., 2020). The *perturbation strength* $k$ is the maximum number of sites which an attacker can perturb. Higher the the Attack Success Rate (ASR), better the attack; the converse holds for F1 score. Our formulation (Eq. 2), solved using two methods – alternate optimization (AO) and joint optimization (JO), along with randomized smoothing (RS), shows a consistent improvement in generating adversarial programs. Differences in ASR, marked in blue, are relative to Baseline. The results on a Java dataset are tabulated in Table 4, Appendix.

computational bench is in Pytorch while the original CODE2SEQ implementation is in TensorFlow, we defer evaluating the performance of our formulation on CODE2SEQ to future work.

**Evaluation metrics.** We report two metrics – Attack Success Rate (ASR) and F1-score. ASR is defined as the percentage of output tokens misclassified by the model on the perturbed input but correctly predicted on the unperturbed input, *i.e.* ASR = $\frac{\sum_{i,j} \mathbb{1}(\theta(\mathbf{x}'_i) \neq y_{ij})}{\sum_{i,j} \mathbb{1}(\theta(\mathbf{x}_i) = y_{ij})}$ for each token $j$ in the expected output of sample $i$. Higher the ASR, better the attack. Unlike (Ramakrishnan et al., 2020), we evaluate our method on those samples in the test-set which were fully, correctly classified by the model. Evaluating on such fully correctly classified samples provides direct evidence of the adversarial effect of the input perturbations (also the model's adversarial robustness) by excluding test samples that have originally been misclassified even without any perturbation. We successfully replicated results from (Ramakrishnan et al., 2020) on the F1-score metric they use, and acknowledge the extensive care they have taken to ensure that their results are reproducible. As reported in Table 2 of (Ramakrishnan et al., 2020), a trained SEQ2SEQ model has an F1-score of $34.3$ evaluated on the entire dataset. We consider just those samples which were correctly classified. The F1-score corresponding to 'No attack' in Table 1 is hence $100$. In all, we perturb 2800 programs in Python and 2300 programs in Java which are correctly classified.

## 5.1 Experiments

We evaluate our work in three ways – first, we evaluate the overall performance of the three approaches we propose – AO, JO, and their combination with RS, to find the best sites and perturbations for a given program. Second, we evaluate the sensitivity of two parameters which control our optimizers – the number of iterations they are evaluated on, and the *perturbation strength* ($k$) of an attacker. Third, we use our formulation to train an adversarially robust SEQ2SEQ model, and evaluate its performance against the attacks we propose.

**Overall attack results.** Table 1 summarizes our overall results. The first row corresponds to the samples not being perturbed at all. The ASR as expected is $0$. The 'Random replace' in row 2 corresponds to both $z$ and $u$ being selected at random. This produces no change in the ASR, suggesting that while obfuscation transformations can potentially deceive ML models, any random transformation will have little effect. It is important to have some principled approach to selecting and applying these transformations.

Ramakrishnan et al. (2020) (referred to as Baseline in Table 1) evaluated their work in two settings. In the first setting, they pick 1 site *at random* and optimally perturb it. They refer to this as $\mathcal{Q}_G^1$. We contrast this by selecting an optimal site through our formulation. We use the same algorithm as theirs to optimally perturb the chosen site *i.e.* to solve for $u$. This allows us to ablate the effect of incorporating and solving the *site-selection* problem in our formulation. In the second related setting, they pick 5 sites at random and optimally perturb them, which they refer to as $\mathcal{Q}_G^5$. In our setup, $\mathcal{Q}_G^1$ and $\mathcal{Q}_G^5$ are equivalent to setting $k = 1$ and $k = 5$ respectively, and picking random sites in $z$ instead of optimal ones. We run AO for 3 iterations, and JO for 10 iterations.

We find that our formulation consistently outperforms the baseline. For $k = 1$, where the attacker can perturb at most 1 site, both AO and JO provide a 3 point improvement in ASR, with

JO marginally performing better than AO. Increasing $k$ improves the ASR across all methods – the attacker now has multiple sites to transform. For $k = 5$, where the attacker has at most 5 sites to transform, we find AO to provide a 6 point improvement in ASR over the baseline, outperforming JO by 1.5 points.

Smoothing the loss function has a marked effect. For $k = 1$, we find smoothing to provide a 10 point increase ($\sim 52\%$ improvement) in ASR over the baseline when applied to AO, while JO+RS provides a 4 point increase. Similarly, for $k = 5$, we find AO+RS to provide a 14 point increase ($\sim 38\%$ improvement), while JO+RS provides an 11 point increase, suggesting the utility of smoothing the landscape to aid optimization.

Overall, we find that accounting for site location in our formulation combined with having a smooth loss function to optimize improves the quality of the generated attacks by nearly 1.5 times over the state-of-the-art attack generation method for programs.

**Effect of solver iterations and perturbation strength $k$.** We evaluate the attack performance (ASR) of our proposed approaches against the number of iterations at $k = 5$ (Figure 4a). For comparison, we also present the performance of BASELINE, which is not sensitive to the number of iterations (consistent with the empirical finding in (Ramakrishnan et al., 2020)), implying its least optimality. Without smoothing, JO takes nearly 10 iterations to reach its local optimal value, whereas AO achieves it using only 3 iterations but with improved optimality (in terms of higher ASR than JO). This supports our hypothesis that AO allows for a much more aggressive exploration of the loss landscape, proving to empirically outperform JO. With smoothing, we find both AO+RS and JO+RS perform better than AO and JO respectively across iterations. We thus recommend using AO+RS with 1-3 iterations as an attack generator to train models that are robust to such adversarial attacks.

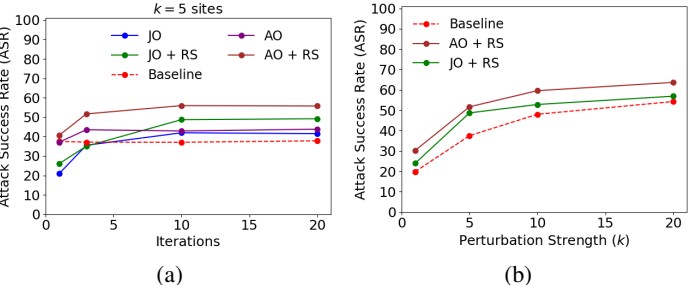

In Figure 4b, we plot the performance of the best performing methods as we vary the attacker's *perturbation strength* ($k$). We make two key observations – First, allowing a few sites ($< 5$) to be perturbed is enough to achieve $80\%$ of the best achievable attack rate. For example, under the AO+RS attack, the ASR is 50 when $k = 5$ and 60 when $k = 20$. From an attacker's perspective, this makes it convenient to effectively attack models of programs without being discovered. 

(a)  (b)

Figure 4: ASRs of our approaches and BASELINE against the number of optimization iterations (4a) and *perturbation strength* of an attacker (4b).

Second, we observe that the best performing methods we propose consistently outperform BASELINE across different $k$. The performance with BASELINE begins to converge only after $k = 10$ sites, suggesting the effectiveness of our attack.

**Improved adversarial training under proposed attack.** Adversarial training (AT) (Madry et al., 2018) is a min-max optimization based training method to improve a model's adversarial robustness. In AT, an attack generator is used as the inner maximization oracle to produce perturbed training examples that are then used in the outer minimization for model training. Using AT, we investigate if our proposed attack generation method (AO+RS) yields an improvement in adversarial robustness (over BASELINE) when it is used to adversarially train the SEQ2SEQ model. We evaluate AT

| Train | Attack (ASR) | | |
|---|---|---|---|
| | BASELINE | AO | AO+RS |
| No AT | 19.87 | 23.16 | 30.25 |
| BASELINE | 17.99 | 18.87 | 19.11 |
| **AO+RS** | **12.73** | **13.01** | **13.75** |

Table 2: We employ an AT setup to train SEQ2SEQ with the attack formulation we propose. Lower the ASR, higher the robustness to adversarial attacks. Training under AO+RS attacks provides best robustness results.

in three settings – 'No AT', corresponding to the regularly trained SEQ2SEQ model (the one used in all the experiments in Table 1), BASELINE - SEQ2SEQ trained under the attack by (Ramakrishnan et al., 2020), and AO+RS - SEQ2SEQ trained under our AO+RS attack. We use three attackers on these models – BASELINE and two of our strongest attacks - AO and AO+RS. The row corresponding to 'No AT' is the same as the entries under $k = 1$ in Table 1. We find AT with BASELINE improves robustness by $\sim$11 points under AO+RS, our strongest attack. However, training with AO+RS provides an improvement of $\sim$16 points. This suggests AO+RS provides better robustness to models when used as an inner maximizer in an AT setting.

## 6 CONCLUSION

In this paper, we propose a general formulation which mathematically defines an adversarial attack on program source code. We model two key aspects in our formalism – location of the transformation, and the specific choice of transformation. We show that the best attack is generated when both these aspects are optimally chosen. Importantly, we identify that the joint optimization problem we set up which models these two aspects is decomposable via alternating optimization. The nature of decomposition enables us to easily and quickly generate adversarial programs. Moreover, we show that a randomized smoothing strategy can further help the optimizer to find better solutions. Eventually, we conduct extensive experiments from both attack and defense perspectives to demonstrate the improvement of our proposal over the state-of-the-art attack generation method.

## 7 ACKNOWLEDGMENT

This work was partially funded by a grant by MIT Quest for Intelligence, and the MIT-IBM AI lab. We thank David Cox for helpful discussions on this work. We also thank Christopher Laibinis for his constant support with computational resources. This work was partially done during an internship by Shashank Srikant at MIT-IBM AI lab.

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

## A   SOLVING THE PAIR OF PROJECTION SUB-PROBLEMS IN EQ. 6

The projection step (5) can formally be described as finding the solution of the convex problem

$$\underset{\mathbf{z}, \mathbf{u}}{\text{minimize}} \quad \|\mathbf{z} - \mathbf{z}^{(t)}\|_2^2 + \sum_i \|\mathbf{u}_i - \mathbf{u}_i^{(t)}\|_2^2$$
$$\text{subject to} \quad \mathbf{1}^T\mathbf{z} \le k, \ \mathbf{z} \in [0,1]^n, \ \mathbf{1}^T\mathbf{u}_i = 1, \ \mathbf{u}_i \in [0,1]^{|\Omega|}, \ \forall i. \tag{9}$$

In Equation 6, we proposed to decompose the projection problem on the combined variables $\mathbf{z}$ and $\mathbf{u}_i$ into the following sub-problems –

$$\underset{\mathbf{z}}{\text{minimize}} \quad \|\mathbf{z} - \mathbf{z}^{(t)}\|_2^2 \qquad \qquad \underset{\mathbf{u}_i}{\text{minimize}} \quad \|\mathbf{u}_i - \mathbf{u}_i^{(t)}\|_2^2$$
$$\text{subject to} \quad \mathbf{1}^T\mathbf{z} \le k, \ \mathbf{z} \in [0,1]^n, \quad \text{and} \quad \text{subject to} \quad \mathbf{1}^T\mathbf{u}_i = 1, \ \mathbf{u}_i \in [0,1]^{|\Omega|}, \qquad \forall i. \tag{10}$$

The following proposition is a closed-form solution which solves them.

**Proposition 1** *Let $\mathbf{z}^{(t+1)}$ and $\{\mathbf{u}_i^{(t+1)}\}$ denote solutions of problems given in (6). Their expressions are given by*

$$\mathbf{z}^{(t+1)} = [\mathbf{z}^{(t)} - \mu\mathbf{1}]_+, \ \text{and } \mu \text{ is the root of } \mathbf{1}^T[\mathbf{z}^{(t)} - \mu\mathbf{1}]_+ = 1; \tag{11}$$

$$\mathbf{u}_i^{(t+1)} = \begin{cases} P_{[\mathbf{0},\mathbf{1}]}[\mathbf{u}_i^{(t)}] & \text{if } \mathbf{1}^T P_{[\mathbf{0},\mathbf{1}]}[\mathbf{u}_i^{(t)}] \le k, \\ P_{[\mathbf{0},\mathbf{1}]}[\mathbf{u}_i^{(t)} - \tau_i\mathbf{1}] & \text{if } \exists \tau_i > 0 \text{ s.t. } \mathbf{1}^T P_{[\mathbf{0},\mathbf{1}]}[\mathbf{u}_i^{(t)} - \tau_i\mathbf{1}] = k, \end{cases} \quad \forall i, \tag{12}$$

*where $[\cdot]_+ = \max\{0, \cdot\}$ denotes the (elementwise) non-negative operation, and $P_{[0,1]}(\cdot)$ is the (elementwise) box projection operation, namely, for a scalar $x$ $P_{[0,1]}(x) = x$ if $x \in [0,1]$, 0 if $x < 0$, and 1 if $x > 1$.*

**Proof.** We first reformulate problems in (6) as problems subject to a single inequality or equality constraint. That is, $\min_{\mathbf{1}^T\mathbf{z} \le k} \|\mathbf{z} - \mathbf{z}^{(t)}\|_2^2 + \mathcal{I}_{[0,1]^n}(\mathbf{z})$ and $\min_{\mathbf{1}^T\mathbf{u}_i = 1} \|\mathbf{u}_i - \mathbf{u}_i^{(t)}\|_2^2 + \mathcal{I}_{\mathbf{u}_i \ge \mathbf{0}}(\mathbf{u}_i)$, where $\mathcal{I}_\mathcal{C}(\mathcal{P})$ denotes an indicator function over the constraint set $\mathcal{C}$ and $\mathcal{I}_\mathcal{C}(\mathcal{P}) = 0$ if $\mathcal{P} \in \mathcal{C}$ and $\infty$ otherwise. We then derive Karush–Kuhn–Tucker (KKT) conditions of the above problems for their optimal solutions; see (Parikh & Boyd, 2014; Xu et al., 2019) for details. □

In (11) and (12), we need to call an internal solver to find the root of a scalar equation, e.g., $\mathbf{1}^T[\mathbf{z}^{(t)} - \mu\mathbf{1}]_+ = 1$ in (11). This can efficiently be accomplished using the bisection method in the logarithmic rate $\mathcal{O}(-\log_2 \epsilon)$ for the solution of $\epsilon$-error tolerance (Boyd et al., 2004). Eventually, the PGD-based JO solver is formed by (4), (5), (11) and (12).

## B Program Transformations

To compare our work against the state-of-the-art (Ramakrishnan et al., 2020), we adopt the transformations introduced in their work and apply our formulation to them. This allows to solve the same setup they try while contrasting the advantages of our formulation.

They implement the following transformations -

- **Renaming local variables.** This is the most common obfuscation operation, where local variables in a scope are renamed to less meaningful names. We use this to replace it with a name which has an adversarial effect on a downstream model.

- **Renaming function parameters.** Similar to renaming variable names, function parameter names are also changeable. We replace names of such parameters with

- **Renaming object fields.** A class' referenced field is renamed in this case. These fields are referenced using `self` and `this` in Python and Java respectively. We assume that the class definitions corresponding to these objects are locally defined and can be changed to reflect these new names.

- **Replacing boolean literals** A boolean literal (True, False) occurring in the program is replaced with an equivalent expression containing optimizable tokens. This results in a statement of the form `<token> == <token>` and `<token> != <token>` for True and False respectively.

- **Inserting print statements.** A `print` with optimizable string arguments are inserted at a location recommended by the *site-selection* variable. See Figure 2 for an example.

- **Adding dead code.** A statement having no consequence to its surrounding code, guarded by an `if`-condition with an optimizable argument, is added to the program.

The last two transformations are *insert* transforms, which introduce new tokens in the original program, whereas the others are *replace* transforms, which modify existing tokens.

They implement two other transformations – inserting a `try-catch` block with an optimizable parameter, and unrolling a while loop once. We did not add these to our final set of transforms since they are very similar to adding print statements and dead code, while producing a negligible effect. Adding every *insert* transformation increases the number of variables to be optimized. Since these two transforms did not impact model performance, we omitted them to keep the number of variables to be optimized bounded.

## C   AN ATTACK EXAMPLE

**Unperturbed**

```python
def __call__(self, *a, **ka):                    Prediction: call
    for key, value in dict(*a, **ka).items(): setattr(self, key, value)
    return self
```

**Random site-selection; Optimal site-perturbation (Ramakrishnan et al., 2020)**

```python
def __call__(self, *a, **ka):                    Prediction: call
    for save, value in dict(*a, **ka).items(): setattr(self, save, value)
    return self
```

**Optimal site-selection; Optimal site-perturbation (This work)**

```python
def __call__(self, *copies, **ka):               Prediction: call
    for key, value in dict(*copies, **ka).items(): setattr(self, key, value)
    return self
```

**Optimal site-selection; Optimal site-perturbation + Smoothing (This work)**

```python
def __call__(self, *datetime, **ka):             Prediction: create
    for key, value in dict(*datetime, **ka).items(): setattr(self, key, value)
    return self
```

Table 3: We present an additional example which contrasts the advantages of different aspects of our formulation. In Figure 1, we saw how selecting an optimal site led to the optimal local variable (`qisrc`) being found. In this example, we show how randomized smoothing, a key solution we propose to ease optimization, helps in finding the best variable. In this case, just finding the optimal site is not enough to flip the classifier's decision (`call`). Smoothing however enables to find a local variable `datetime` which flips the classifier's decision to `create`.

# D    ADDITIONAL RESULTS

## D.1    JAVA DATASET

We tabulate results of evaluating our formulation on an additional dataset containing Java programs. This dataset was released by Alon et al. (2018) in their work on CODE2SEQ. The transformations were implemented using Spoon (Pawlak et al., 2016). We find the results to be consistent with the results on the Python dataset. AO + RS provides the best attack.

| Method | $k = 1$ site | | | | $k = 5$ sites | | | |
|---|---|---|---|---|---|---|---|---|
| | ASR | | F1 | | ASR | | F1 | |
| No attack | 0.00 | | 100.00 | | 0.00 | | 100.00 | |
| Random replace | 0.00 | | 100.00 | | 0.00 | | 100.00 | |
| BASELINE* | 22.93 | | 70.75 | | 33.16 | | 59.45 | |
| AO | 25.95 | +3.02 ▲ | 67.17 | -3.58 ▲ | 33.41 | +0.25 ▲ | 59.03 | -0.42 ▲ |
| JO | 23.26 | +0.33 ▲ | 70.08 | -0.67 ▲ | 33.65 | +0.49 ▲ | 58.85 | -0.60 ▲ |
| AO + RS | 29.08 | +6.15 ▲ | 63.90 | -6.85 ▲ | 40.53 | +7.37 ▲ | 51.91 | -7.54 ▲ |
| JO + RS | 26.71 | +3.78 ▲ | 65.41 | -5.34 ▲ | 38.30 | +5.14 ▲ | 53.14 | -6.31 ▲ |

Table 4: Performance of our formulation on a dataset containing Java programs (Alon et al., 2018). See Table 1 for results on a Python dataset.

## D.2    FALSE POSITIVE RATE (FPR) UNDER DIFFERENT ATTACKS

We evaluated the False Positive Rate (FPR) of the model when provided perturbed programs as inputs. It is important the perturbations made to the programs cause the model to start predicting ground truth tokens instead of incorrect tokens. In principle, our optimization objective strictly picks replacements which degrade the classifier's performance. As a consequence, we should expect that the model does not end up perturbing the program in a way which leads it closer to the ground truth. We empirically find the FPR consistent with this explanation. As seen in Table 5, in all our attacks, the FPR is almost 0, validating that our perturbations do not introduce changes which result in the model predicting the right output.

| Method | **FPR**, $k$=1 | **FPR**, $k$=5 |
|---|---|---|
| No attack | 0.0000 | 0.0000 |
| BASELINE | 0.0077 | 0.0130 |
| AO | 0.0073 | 0.0114 |
| AO+RS | 0.0075 | 0.0148 |

Table 5: False positive rates of the model under different attacks.

## D.3    EFFECT OF VARIOUS TRANSFORMATIONS

We study the effect of different transformations used in our work on the attack success rate of our attacks. In our analysis, we found that omitting `print` statements do not affect the ASR of our attacks. This is helpful since for the current task we evaluate (generating program summaries), a `print` statement likely affects the semantics of the task.

| Training | Model's F1-score | ASR, $k$=1 | | | ASR, $k$=5 | | |
|---|---|---|---|---|---|---|---|
| | | BASELINE | AO | AO+RS | BASELINE | AO | AO+RS |
| All transformations (All) | 33.74 | 19.87 | 23.16 | 30.25 | 37.50 | 43.53 | 51.68 |
| (All − `print`) | 33.14 | 16.39 | 20.22 | 26.46 | 35.14 | 42.70 | 51.24 |
| (All − vars, function params) | 30.90 | 21.48 | 26.36 | 31.43 | 37.21 | 37.53 | 48.54 |

Table 6: The effect of variable names, function parameter names, and print statements on the robustness of the learned model. These results correspond to the Python dataset.

We further investigated the effect of variable names and function parameter names on our attacks. The SEQ2SEQ model and the task of generating program summaries can appear to overly rely on the presence of variable and parameter names in the program. To empirically ascertain this, we masked

out all the variable and parameter names from our training set and trained the model with place-holder tokens. We find the model's performance to remain similar (row `All - vars, function params`: 1, Table 6). Likewise, the attack performance also remains similar. We test another condition where we mask all these token names in the test set as well, during inference. We find a decrease in performance (row 4, Table 6) under attack strength $k = 1$, but the trend observed remains – AO+RS $>$AO $>$BASELINE. The ASR is largely unchanged under $k = 5$.

