# OpenReview forum: "Generating Adversarial Computer Programs using Optimized Obfuscations"
_ICLR.cc/2021/Conference — ICLR 2021 Poster_

### Official Review · AnonReviewer3 · 2020-10-24
**Adversarial code mutations by jointly choosing mutation site and type**

**Rating:** 6
**Confidence:** 4

**Review:**

# Summary

This work tackles the problem of adversarial attacks against ML models for code-understanding tasks, such as function summarization. It formulates the problem as the adversarial application of existing semantics-preserving program transformations (e.g., renaming variables), by jointly optimizing on the location of such a transformation, and the argument to the transformation (e.g., what to replace an existing identifier with). It shows that such adversarial examples increase the attack success rate over baseline approaches, and training with such examples increases the robustness of the resulting model to the same or baseline attacks.

# Strengths:
1. Clear description of the mathematical formulation and the approach.
1. Results over the baselines and seq2seq model seem to be very promising.

# Weaknesses:
1. The core model used is not a state-of-the-art model for code understanding. Transformer or GGNN-based models have been shown to do much better on code-understanding tasks than seq2seq models.
1. The choice of task and perturbation actions are expedient to try out the approach, but practically questionable (see below).
1. The choice of model on which to evaluate the approach is poor; seq2seq is a weak baseline for function-naming tasks.


# Feedback

1. As an application of techniques for adversarial robustness to code-understanding tasks, this paper is a great read. I enjoyed the description of the approach (although I'm not an expert on adversarial examples), and found the direction intuitive and well presented. I enjoyed reading your paper.

1. However, digging below the surface, I'm uneasy with several aspects of the concrete realization of your approach.

1. Perturbations must be semantics preserving. Although most of the perturbations you describe have that property, inserting `print` statements does not. Changing the output of a program constitutes a major modification of semantics, unless you further limit, precisely, your definition of "semantics preserving". As an obvious example, a function
```
def ___(a):
    result = a*2
    return result
```
is very similar to
```
def ___(a):
    result = a*2
    print(result)
    return result
```
however I could argue that first should be named `compute_double` while the second should be named `compute_and_print_double`. I would include in this category logging statements as well; they alter the program semantics. What might not are operations that have no side effects (as far as inputs/outputs/state are concerned), for instance writing to `/dev/null`

1. I'm also a little skeptical about renaming local variables and function parameters. Those may have an inordinate influence on the particular model and task you chose. However, some work has found that tokens themselves don't affect the results of some models. For example, in `Allamanis, Miltiadis, Marc Brockschmidt, and Mahmoud Khademi. "Learning to represent programs with graphs." arXiv preprint arXiv:1711.00740 (2017).`, having token labels makes no difference when solving a variable-misuse task, but actually helps when solving variable naming (see Table 2).

1. What's more, there's some evidence that completely removing identifiers from a program (interestingly enough, to obfuscate it) may not destroy the relevant information needed to recover the identifiers. See, for example `Bichsel, Benjamin, Veselin Raychev, Petar Tsankov, and Martin Vechev. "Statistical deobfuscation of android applications." In Proceedings of the 2016 ACM SIGSAC Conference on Computer and Communications Security, pp. 343-355. 2016.` which recovers missing identifiers. In principle, that tells me that useful identifiers may still be recoverable from completely obfuscated programs from contextual structure. So this leads me to believe that what you present here is more about a model that wasn't meant to understand relational properties of its input rather focusing instead on tokens alone, and less about an inherent limitation in how we build neural models for code.

1. Even ignoring the above, I'd want to see more about how imperceptibility applies to the task at hand. Your attacks may well change a predicted name from its original to something else, but how far away is that new prediction, and does it matter? When classifying, switching classes is an important effect of attack. However, when predicting a description, changing predicted tokens doesn't necessarily change the meaning of the description. What kind of change constitutes a successful attack and what kind of change doesn't disturb the correctness of the prediction? As a perhaps naive example, if you attacked a function to change its name from `read_file_from_disk`  to `collect_contents_from_storage`, although a huge change in tokens involved, I'd say the attack was unsuccessful.

1. Finally, to lend credence to your results, I would want to see your approach applied to state-of-the-art models for program understanding. seq2seq is not state of the art. For variable-name prediction, something like `Alon, Uri, Shaked Brody, Omer Levy, and Eran Yahav. "code2seq: Generating sequences from structured representations of code." arXiv preprint arXiv:1808.01400 (2018).` might be more appropriate.

1. All in all, I learned much about adversarial training from reading your paper (since I hadn't read much about it before). However, I don't find that your results will change how models of code are built, because they are evaluated on what is now considered a fairly weak model basis, and the criteria for what constitutes a successful attack are not well defined and appear overly pessimistic.

---

> ### Author Response · Authors · 2020-11-17
> **Response to AnonReviewer 3 - Part 2**
>
>
> 3. **Feedback #6 - Effect of transformations.**
>
>  Interpreting by how much a program changes on transforming it is a great question. As you suggest, it is not enough if the predicted tokens are different from ground truth - they need to be semantically different as well. We observe the following
> - If our aim is solely to construct an attack such that the resulting prediction is semantically very different from the predicted sequence/ground truth, we could always adopt a _targeted attack_ setting.  Currently, we add these transformations in a way to make a prediction such that it is _as different as possible from the original_. This difference is measured by the change in the model's loss. In the targeted attack setting however, the loss function can be changed in a way to find a transformation which moves the prediction to **a specific sequence of tokens chosen by the attacker**. This would allow us to predict any output function name of our liking by changing the loss function to something like the Carlini-Wagner loss from the current cross entropy loss. This is a known setting for generating targeted adversarial attacks; see Carlini & Wagner, 2017 [1] for details. We plan to add the evaluation of targeted attacks in the revision.
> - The disadvantage of a targeted attack is that it cannot be used to robustly train an existing model (sub-section _Adversarial training_ in section 5.2 in our current draft). We hence did not explore the targeted setting in our current work.
> - Nevertheless, we are currently analyzing the semantic similarity between the outputs of the unperturbed programs and their perturbed programs, to show how effective these adversarial perturbations are in the untargeted setting.
>
>     [1] Carlini, N., & Wagner, D. (2017, May). Towards evaluating the robustness of neural networks. In 2017 ieee symposium on security and privacy (sp) (pp. 39-57). IEEE.
>
>
> 4. **Feedback #7 - State-of-the-art model.**
>
> Thank you for this suggestion. The reason we had not evaluated `code2seq` initially was because .a. `code2seq` and `seq2seq` had similar F1 scores on this task (see the baseline work, Ramakrishnan _et al._, 2020), and .b. Implementing our framework was relatively easier on `seq2seq`. We are now in the process of evaluating `code2seq`, and will update our results soon to this response. `code2seq` was evaluated previously against `code2vec`, bi-LSTMs, and Transformers, and outperformed all of them. Thus, modeling our task with `code2seq` should represent the state-of-the-art.
>
> We further evaluated our framework on an additional Java dataset, and found it to similarly outperform the baseline work. We summarize results of adversarial programs under the site perturbation strength  $k=1$ below. As we can see, our proposals (AO and AO+RS) outperform Baseline and yield comparable ASR results across the dataset choices. We provide details in Appendix D of our revised draft.
>
> Attack  | ASR - Java (new) | ASR - Python (from submitted draft)
> ------------- | ------------- | -------------
> No attack  |  0 | 0
> Random replace  | 0 | 0
> Baseline (Ramakrishnan _et al._, 2020)  | 22.93 | 19.87
> AO | **25.95**  | 23.16
> AO + RS  | **29.08** | 30.25
>
>
> 5. **Feedback #8, How our results will change how models of code are built.** Although we do not discuss this in our current draft, we have continued this line of research to address the very concern you raise. We are in the middle of reformulating such models' loss functions in a way to show that for a given model choice, it is possible to improve the model's ability to learn general features of programs by adopting ideas presented in this work. This investigation merits another study in itself, and thus we exclude it from the scope of the current formulation we propose.
>
> 6. **Releasing source code.** The anonymized version of our codebase, which we missed adding to our draft, can be found at https://anonymous.4open.science/r/a2ff567d-0e7e-476e-b80e-3849eddeb729/
>
> We hope our response has addressed most of your concerns, and we hope it highlights our efforts in bridging the adversarial ML community and the PL/SE community. We are glad to continue a discussion to address any other questions you may have, and in the process conduct additional experiments.

---

> ### Author Response · Authors · 2020-11-17
> **Response to AnonReviewer 3 - Part 1**
>
> Thank you for the detailed and very insightful comments. We’re very glad you enjoyed reading our work, and likewise, we found the set of perceptive questions you raise in your feedback very insightful, pushing us to think of a tighter experiment design. We currently have the following additional experiments underway, some of which we have results for -
> - **An additional dataset.** We have results on a Java dataset.
> - **State-of-the-art `code2seq` model.** We are in the process of evaluating our work on `code2seq`
> - **Analysis of effect of transformations.** We ran additional experiments to support our comments on your feedback points #2, #3, #4, #5.
>
> We address each comment you shared in the 'Feedback' section of your response.
>
> 1. **Feedback #2 - `print` statements**.
>
> We agree with your observation that in the general setting, printing to the console or to a log affects the choice of a descriptive name provided to that functionality, thus changing the semantics. In the context of our current work, we assume that the programs we deal with are production systems and any output to the console would be redirected to `/dev/null/` by default, or would be unmanned, and that any output that needs to be recorded will be formally logged by a logging utility.
> Nevertheless, we verified our dataset for occurrence of `print` commands and found the following -- $1.34$% of the programs in the Python dataset contained a `print` statement ($0.12$% in Java). This small number of occurrences suggests such production systems do not have `print` statements routinely present in them. However, investigating further, we found that all of these samples containing `print` statements had the word `print` in their function names as well. This validates your observation that the presence of print operations affects how functions are named, thus affecting this task. We excluded this transformation from our overall suite of transformations, and the AO+RS attack’s ASR dropped negligibly from 30.25 to 28.8 for perturbation strength $k=1$. We will update our draft with this information.
>
> On that note, being agnostic to specific threat models, our formulation provides the flexibility to seamlessly add and drop transformations. If printing to the console is not permissible in a specific threat model, one can always exclude it from the list of transformations.
>
> 2. **Feedback #3, #4, #5 - Effect of variable and function parameter names.**
>
> These are great observations! The extent to which models attend to structural information (ASTs, program graphs) vs. syntactic information (variable names, tokens) is pertinent to answering whether some of our transformations conveniently favor a task which depends solely on syntactic information. **We show that the model in our task does not solely depend on syntactic information**, which suggests that the variable and parameter rename transformations are not expedient to the task we evaluate. We investigated how our task+model choice responded to variable names in the following manner --
> - We masked all the variable names in the programs in the train set and re-trained the model. Doing so allowed us to measure how much of the model's performance relied on information available in variable names. The F1-score dropped from $33.74$ in the unmasked setting to $33.14$ in the masked setting for Python. Further, we masked variable names + function parameter names, and the accuracy dropped only to $30.9$, while masking all _replace_ transformations dropped the accuracy to $30.8$. This suggests that the model does not rely solely on variable and function parameter names to learn program summaries, and instead learns the structural properties in these programs.
> - We attacked this masked model with the different settings we discuss in our work and find the attack success rate (ASR) to remain largely unchanged. Thus, changing variable and parameter names in a program and passing it to a model which has not seen any such names during training remains a good choice for an adversarial attack.
> - Further, we analyzed the optimal transformations selected by the optimization process in these perturbed programs, to confirm whether these replace transforms indeed accounted for the unchanged ASR that we see in this masked setting above. One possibility is that on attacking the masked model, the optimization could instead prefer _insert_ transforms, compensating for the ineffectiveness of _replace_ transforms. *We found that this is not the case*. Variable and parameter name changes accounted for $51$% of the overall optimal transformations selected by our algorithm in the unmasked setting, and accounted for $48$%, nearly unchanged, in the masked setting.
> This confirms that the specific task we investigate in this work is not expedient to the set of transformations we propose to perturb programs with, and to the formulation we propose.
> We will update these results in the appendix of our draft in the coming week.

---

### Official Review · AnonReviewer4 · 2020-10-27
**ICLR 2021 Review Conference Paper1336**

**Rating:** 7
**Confidence:** 4

**Review:**

Summary:
* This paper proposes an optimization problem to adopt insert/replace operations (program obfuscations) to generate adversarial programs. They apply it to the task of program summarization, and show that they outperform the existing baseline published in 2020. In particular, one of the main contributions is the identification of the site-perturbation and site-selection process, and formalizing them as practical optimization based on PGD.

Strengths:
* The problem is relevant and challenging
* Formalization is elegant and clear
* Contributions and related works are relevant and significant
* Experimental comparison against relevant, recent work

Weaknesses:
* The task of program summarization is somehow sinmple
* It is not well-motivated why to limit the perturbation strength k, since we are talking about a program (see also discussion in [2])
* Attack success rate improvement over baseline is moderate.
* There seems to be no plan for the authors to publicly release the code.

Detailed comments:
* The intuition of using site-selection and site-perturbation is extremely interesting and is well developed. The formalization of the problem is very elegant, and is convincing. Modeling the “location” of the program modification and identifying that this increases attack success rate is a very important contribution.
* Related work is generally relevant and significant. I also strongly appreciate that there is an experimental comparison against a recent paper from 2020, representing the state of the art. As related work, I would also recommend the authors to have a look at, which seems highly related from the security domain (see [1,2,3] below).
* I think the task of program summarization is fairly simple, and it would have been interesting to see something a bit more complex. Some “replace” operations such as renaming of variables seems to be very related to the attack in [1]. The “insertion” operation seems to be quite trivial, such as adding prints or dead code. This could be easily defeated via preprocessing commonly done by compiler’s program analysis, such as unreachable code elimination (see problem-space constraint [2]).
* Nevertheless, I do believe that the paper proposes a very interesting perspective. I would also encourage the authors to consider releasing the source code.
* Minor comments:
    * Typo “reproducability” on page 6

References:
* [1] Quiring et al. "Misleading authorship attribution of source code using adversarial learning.” USENIX Security Symposium, 2019.
* [2] Pierazzi et al, “Intriguing Properties of Adversarial ML Attacks in the Problem Space”, IEEE Symp. S&P 2020
* [3] Demetrio, Luca, et al. "Adversarial EXEmples: A Survey and Experimental Evaluation of Practical Attacks on Machine Learning for Windows Malware Detection." arXiv preprint arXiv:2008.07125 (2020).

---

> ### Author Response · Authors · 2020-11-17
> **Response to AnonReviewer 4 - Part 2**
>
>
> 3. **Need for perturbation constraint $k$.** We think the following conditions should encourage accounting for $k$ in any formulation solving this problem -
> - **Human in the loop.** We account for a possible threat model where a human can likely scan a perturbed program. For such a use-case, we would want to ensure that a minimal number of constructs in a program changes as a consequence of perturbing it. For instance, consider renaming variables in a program containing multiple local variables, and spanning multiple lines. It is quite likely  for a human expert carefully scanning such a program to ignore 1-2 variable names not fitting the context of the program.
> - **Poor generalization.** A controllable $k$ is also preferred for adversarial training (AT) that we studied in Sec. 5.2  . Consider obfuscating all possible sites in a program, i.e., $k=N$, where $N$ is the number of possible sites in a program. Such fully obfuscated programs, when used to train a robust model through the AT algorithm, would likely perform poorly on an unattacked test set. This is because such models would train to expect almost every line to have instances of _insert_ or _replace_ transformations. Controlling the extent of obfuscation through $k$ would help avoid this situation, thus improving generalization. This has been known as the robustness-accuracy tradeoff in the adversarial ML literature [6,7,8].
>
>     [6] Tsipras, Dimitris, et al. "Robustness may be at odds with accuracy." arXiv preprint arXiv:1805.12152 (2018).
>
>     [7] Su, Dong, et al. "Is Robustness the Cost of Accuracy?--A Comprehensive Study on the Robustness of 18 Deep Image Classification Models." Proceedings of the European Conference on Computer Vision (ECCV). 2018.
>
>     [8] Zhang, Hongyang, et al. "Theoretically principled trade-off between robustness and accuracy." arXiv preprint arXiv:1901.08573 (2019).
>
> 4. **Current _replace_ and _insert_ transformations.** Regarding your valid concern on the efficacy of the _replace_ transforms, `print` statements, and dead code, we make some general observations followed by a note on why they pose a serious threat despite being theoretically easy to detect them through a pre-processing step --
> - The replace transforms mentioned in the reference Quiring et al., 2019 you suggested are indeed similar to the ones we propose. They focus on replacing variable names to match programmer style, while we pose no such additional constraints. We instead propose a general set of such _replace_ obfuscation transformations, and provide a framework to evaluate them.
> - The importance and relevance of _insert_ transformations depend on the threat model being considered. We show how this class of transformations, independent of any specific such threat model, can be incorporated in a general formulation like the one we present. We show that if such transformations are allowed, _insert_ transforms can pose a serious risk.
> - Specifically, a realistic threat model can have all the console generated output from a program directed to `/dev/null/`, or suppressed by a unit-test framework. This is the standard setting for programs in production. At best, such production systems use loggers to record outputs to a file. In such a setting, inserting `print` statements with carefully chosen arguments can prove to be very effective.
> - You rightly identify dead code elimination to be an effective pre-processing step in the context we introduce. However, a surprisingly simple number of evaluation conditions fail to be detected by state-of-the-art dead code elimination (DCE) analyzers. Such DCE analyzers use SMT solvers to determine the validity of conditional statements [9]. Most SMT solvers do not support trigonometric functions, and transcendental numbers like $\pi$ and $e$. The DCE system in an industrial-grade static analyzer like Clang/LLVM fails to detect a condition like
>     ```
>     if np.pi/2 < -1:
>         do something
>     ```
> to always be false. Several such simple conditions, when used in `if` statements, will not be detected by state-of-the-art DCE analyzers, thus providing a valid way to introduce new code which does not alter the semantics of the original program.
>
>     [9] Sasnauskas, Raimondas, et al. "Souper: A synthesizing superoptimizer." arXiv preprint arXiv:1711.04422 (2017).
>
> 5. **Releasing code.** Thank you for spotting this. Yes, we definitely plan to release all our source code. The anonymized version, which we missed adding to our draft, can be found at https://anonymous.4open.science/r/a2ff567d-0e7e-476e-b80e-3849eddeb729/

---

> ### Author Response · Authors · 2020-11-17
> **Response to AnonReviewer 4 - Part 1**
>
> Thank you for your feedback and the very pertinent recent works you suggested. It is great that you find our formulation clear and elegant. We also appreciate that you find our experiment results strong, and our comparison against a state-of-the-art work a strength.
>
> We address the comments and feedback you provided --
>
> 1. **Program summarization is a simple task.** Thank you for your observation on the difficulty of the chosen task. We believe that this specific task is in fact quite challenging for the software engineering (SE) community. This task requires inferring a short, meaningful natural language description from 10-30 lines of programs in a function body. This requires reasoning over both syntactic (tokens, variable names) and structural (ASTs, program graphs) properties of the program. What also adds to the difficulty of this task is that unlike other tasks like code attribution and malware detection, this is *not an N-class classification task, but rather a sequence prediction task*.
> Over the last three years, the PL/SE community has consistently used this task to help benchmark different code representations and models  [1,2,3,4,5]. To evaluate the effect on a stronger model on this task, we are currently in the process of additionally evaluating code2seq [6], the state-of-the-art model for programs. We will update our results soon.
> That said, thank you so much for bringing to our attention very relevant works from the security community. We discuss this in our following note.
>
>     [1] Allamanis, Miltiadis, et al. "Learning to represent programs with graphs.", ICLR 2017.
>
>     [2] Alon, Uri, et al. "A general path-based representation for predicting program properties.", PLDI 2018.
>
>     [3] David, Yaniv, et al. "Neural reverse engineering of stripped binaries.", OOPSLA 2020.
>
>     [4] Jain, Paras, et al. "Contrastive Code Representation Learning." arXiv preprint arXiv:2007.04973 (2020).
>
>     [5] Wang, Yu, et al. "Learning Semantic Program Embeddings with Graph Interval Neural Network.", OOPSLA, 2020.
>
>     [6] Alon, Uri, et al. "code2seq: Generating sequences from structured representations of code.", POPL, 2018.
>
> 2. **Related work.** Thank you for pointing out these very relevant works. We are very pleased to learn of these efforts happening in the security community. Having reviewed it, we will definitely add it to the section on related work in our draft. Inspired by examples of malware classification tasks presented in these references, we are in the process of evaluating our framework on a malware classification task. We also malware classifiers in our introduction, while motivating this problem. Time permitting, we plan to have results for it by the end of this discussion period. A note about each of the three referenced works follows --
> - Quiring et al., 2019 -- The task of code attribution is very similar to program summarization. This task too requires a code snippet to be analyzed by a model, and accounting for different syntactic and structural (ASTs, program graphs) components in the snippet, infer its authorship. They discuss a number of obfuscation operations, all of which can easily be accommodated in the framework we propose.
> A note on their dataset -- these programs are sourced from a programming contest like Google Code Jam. In our experience, we find these datasets to represent very constrained conditions in which programs are written. The PL/SE community has hence attempted to perform similar tasks on production-grade code available on platforms like GitHub. Our task deals with one such dataset.
> - Pierazzi et al., 2020 -- This was a great read. They make very similar observations regarding the nature of these tasks. We are very motivated to eventually extend our work to the task of malware detection, and highlight the potential of such adversarial perturbations. This task has a lot in common with our program summarization task -- a model is expected to parse and learn properties in a blurb of code and classify it as being malicious or not. A key difference though is in their task, the input features to the state-of-the-art ML model they benchmark against are all hand-crafted components, picked from the source and execution traces of the input program. Our task setting involves just the raw source code itself input to the ML model, thus providing no information on what features might affect the model the most. This makes our setting a general one to adversarially perturb.
> - Demetrio et al., 2020 -- This is again a very interesting work which evaluates the task of portable executables. We see ourselves naturally transitioning to this flavor of tasks, but our current focus is to demonstrate the adversarial effect on models which accept raw source codes as inputs and try to statically infer properties in them.

---

### Official Review · AnonReviewer1 · 2020-10-28
**Generating Adversarial Computer Programs using Optimized Obfuscations**

**Rating:** 6
**Confidence:** 3

**Review:**

Summary:
This paper presents a principled way of applying obfuscation transformation to create adversarial programs. A general mathematical formulation is presented to model the site locations and the transformation choice for each location in a perturbed program. Then, a set of first-order optimization algorithms are used to solve the proposed formulation.

Reasons for score:
The central contribution of this work is the addition of a formal approach to select perturbation locations along with the types of transformations for adversarial program generation. The mathematical formulation enables a principled approach to achieve this. However, the overall efficacy of the proposed approach over the state-of-the are demonstrated using only one model and dataset. Also, the evaluation method could use some clarification (Please see the weakness section below). I believe a more elaborate analysis will strengthen the claim and make the incremental contribution more generalizable.

Strength
1. The formulations for site-selection, site-perturbation, and perturbation strength provides a principled way of generating adversarial programs by casting it as a constrained optimization problem. It focuses on the problem of finding both optimal location and optimal transformation whereas the state-of-the-art addresses only the latter.
2. The proposed joint optimization improves attack success rate compared to the state-of-the-art approach.
3.  Ablation study is performed to demonstrate  the effectiveness of randomized smoothing.

Weakness
1. The comparative study would be more convincing with evaluation results from more than one model architecture and dataset.
2. Evaluation is performed only on fully correctly classified samples. Although this approach provides a good way of understanding the perturbation effects on original positive examples, any impact on original negative examples is not visible from these experiments. I think the change in the false positive rates from the original model results could be useful to understand the impact of the generated obfuscations.


Questions to author: (Please address the cons above)
1. Yefet et al. (2019) generates adversarial examples for “targeted attacks”. How effective is the proposed approach for  a “targeted attack”?

---

> ### Author Response · Authors · 2020-11-17
> **Response to AnonReviewer 1**
>
> We thank you for your feedback. It is great that you consider our work to be well founded and principled. We address each of the points you raise in your evaluation.
>
> 1. **Models and datasets.**
>  In the paper, we evaluated our proposal on the program summarization task, a task which has been evaluated by multiple works in the recent past to showcase the effectiveness of ML models for code. However, our formulation and optimization approach is agnostic to the downstream task, properties of a dataset, optimizer used, and the choice of a loss function used for the downstream task. To demonstrate this, we add the following experiments --
>
>     1.1. **New dataset: Java**.
> In addition to evaluating our formulation on Python, we also evaluated it on another dataset of Java programs released by Alon _et al._ [1].  The transformations were implemented using  _Spoon_ [2], a library which allows editing ASTs of Java programs. We summarize results of adversarial programs under the site perturbation strength  $k=1$ below. As we can see, our proposals (AO and AO+RS) outperform Baseline and yield comparable ASR results across the dataset choices. We provide details in Appendix D of our revised draft.
>
>     Attack  | ASR - Java (new) | ASR - Python (from submitted draft)
>     ------------- | ------------- | -------------
>     No attack  |  0 | 0
>     Random replace  | 0 | 0
>     Baseline (Ramakrishnan _et al._, 2020)  | 22.93 | 19.87
>     AO | **25.95**  | 23.16
>     AO + RS  | **29.08** | 30.25
>
>     [1] Alon, Uri, et al. "code2seq: Generating sequences from structured representations of code." arXiv preprint arXiv:1808.01400 (2018).
>
>     [2] Pawlak, Renaud, et al. "Spoon: A library for implementing analyses and transformations of java source code." Software: Practice and Experience 46.9 (2016): 1155-1179.
>
>     1.2. **New model architecture: `code2seq`.** Thanks for the suggestion. We are in the process of evaluating another model `code2seq`, and will update our results soon to this response. The rationale behind choosing `code2seq` as the additional model is that Alon _et al._ [1] compared `code2seq` with `code2vec`, bi-LSTMs, and Transformers, and found it to outperform all of them. Thus, modeling our task with `code2seq` should represent the state-of-the-art.
>
> 2. **Evaluating the FPR.** Very good question! It is indeed interesting to see how optimized adversarial obfuscations affect negative samples. We are in the process of re-evaluating our results with FPR as a metric, and will update our results soon to this response.
>
> 3. **Targeted attacks.** Thank you for pointing out targeted attacks. Before we address it, we further clarify why the untargeted attack is the focus of our work.
>
>     First, we lay out our work with the end goal of providing a credible defense to the attacks that we propose, and not confine our work to just demonstrating a successful attack strategy. Thus, when a robust adversarial training (AT) strategy (Madry et al., 2018) is applied, an untargeted attack is favored by maximizing the Cross Entropy (CE) loss.  As a result, we employ the untargeted setting in both the attack and defense sections of our work.
>
>     Second, the choice of an attack loss function matters when designing targeted attacks. For targeted attacks, the CE loss is not an appropriate choice. Instead, C&W loss [3] becomes more effective. However, switching over a loss function will not allow for a fair comparison with our baseline Ramakrishnan _et al._, 2020.
>
>     We admit that your question on targeted attacks is valuable, and we will conduct additional experiments and add the performance of targeted attacks in the supplement. However, due to our limited computing resources and the requested other experiments, we may consider the generation of targeted attacks as a low priority in the rebuttal phase (we will try to make it, and will definitely add it after rebuttal).
>
>     [3] Carlini, N., & Wagner, D. (2017, May). Towards evaluating the robustness of neural networks. In 2017 ieee symposium on security and privacy (sp) (pp. 39-57). IEEE.
>
> We hope our response has addressed most of your concerns, and we hope it highlights our efforts in bridging the adversarial ML community and the PL/SE community. We are glad to continue a discussion to address any other questions you may have, and in the process conduct additional experiments.

---

### Decision · Program_Chairs · 2021-01-07
**Final Decision**

**Decision:**

Accept (Poster)

**Comment:**

This is a nice paper on generating adversarial programs. The approach is to carefully use program obfuscators. After discussion and improvements, reviewers were generally satisfied with the approach and evaluation. The problem domain was also found to be of interest.